# Superficial Venous Reflux Intervention Guided by Triggered Angiography Non-Contrast-Enhanced Sequence Magnetic Resonance Imaging: Different QFlow Pattern from Health Controls

**DOI:** 10.3390/jpm11080751

**Published:** 2021-07-30

**Authors:** Chien-Wei Chen, Yuan-Hsi Tseng, Yueh-Fu Fang, Min Yi Wong, Yu-Hui Lin, Yao-Kuang Huang

**Affiliations:** 1Department of Diagnostic Radiology, Chia-Yi Gung Memorial Hospital and Chang Gung University, College of Medicine, Taoyuan 333, Taiwan; chienwei33@gmail.com; 2Division of Thoracic and Cardiovascular Surgery, Chia-Yi Chan Gung Memorial Hospital and Chang Gung University, College of Medicine, Taoyuan 333, Taiwan; 8802003@cgmh.org.tw (Y.-H.T.); mynyy001@gmail.com (M.Y.W.); vw200162@gmail.com (Y.-H.L.); 3Department of Thoracic Medicine, Linkou Chang Gung Memorial Hospital and Chang Gung University, College of Medicine, Taoyuan 333, Taiwan; dr.fang.yf@gmail.com

**Keywords:** MRI, personalized, endovascular, QFlow, varicose, reflux

## Abstract

(1) Background: To assess the effectiveness of triggered angiography non-contrast-enhanced (TRANCE)-magnetic resonance imaging (MRI) in superficial venous reflux and its difference from health controls. (2) Methods: Thirty patients underwent TRANCE MRI before surgical intervention of their superficial venous reflux of the legs. Ten healthy volunteers were included as a control. (3) Results: TRANCE MRI involves the major tributaries, thus enhances the additional ablations in 20% of patients. QFlow pattern of superficial venous reflux (QFlow GSV/PV MF ratio > 1) was compared with the duplex scan (SFJ reflux) using Cohen’s kappa coefficient at 0.967. The 30 morbid legs undergoing TRANCE MRI-guide interventions and the healthy volunteers’ legs on the same side were compared. The stroke volumes (SV) are higher in EIV (*p* = 0.021) in the left-leg-intervention group. The mean flux (MF) is higher in the EIV (*p* = 0.012) and trend of increasing in GSV segment (*p* = 0.087) in the left-leg-intervention group. The QFlow of 10 patients with right leg intervention are higher in GSV in the right-leg-intervention group (SV *p* = 0.002; FFV *p* = 0.001; MF *p* = 0.001). QFlow data is shown for all legs for superficial venous intervention with GSV/PV (MF) ratio > 1. (4) Conclusions: Typical figures in QFlow (GSV/PV MF ratio > 1) could be observed in the morbid limbs but not in the controls.

## 1. Introduction

Venous diseases of the lower extremities include minor varicose veins and static ulcers, ranging from ambulatory venous hypertension, vascular compression (May Thurner syndrome) to potentially fatal status (such as deep vein thrombosis plus pulmonary emboli) [1,2,3,4,5,6]. Only a few modalities are available for objective venous evaluation of the lower limbs. The venous system is not precisely enhanced on the computed tomography (CT) venogram, and high-quality enhancement requires specific access (from a morbid limb). Compared with conventional angiography, most magnetic resonance venography (MRV) techniques involving contrast media have exhibited higher sensitivity in detecting lesions in vessels [7]. The triggered angiography non-contrast-enhanced (TRANCE) technique records differences in the vascular signal intensity during the cardiac cycle for subsequent image subtraction and provides a vascular image without requiring contrast agents. The clinical application of this technique has enabled the evaluation of the anatomical structure of the whole venous system in the lower extremities [6,8,9]. TRANCE-MRI reveals the location of not only venous compression but also all major collateral veins, thus helping to achieve superior venous ablation results. The interventions of the superficial venous reflux could be well discussed and be personalized according to the patients’ preference. We integrated this technique into surgical planning for superficial venous reflux of the legs and summarized the value of this protocol (Appendix A).

## 2. Materials and Methods

### 2.1. Patients

The Institutional Review Board (IRB) of Chang Gung Memorial Hospital approved this study (IRB number: 201802137B0, 202001213B0 and 202100938B0), which included consecutive patients who received TRANCE-MRI for the evaluation of the venous diseases of their lower extremities at a tertiary hospital between April 2017 and May 2021. We prospectively collected and retrospectively analyzed their data to determine their clinical significance. All patients were suspected to have venous problems in their lower extremities. Patients were excluded if they exhibited poor compliance or had multiple comorbidities that prevented them from lying down for the whole TRANCE-MRI exam. Initially, 230 patients underwent TRANCE-MRI for the venous examination of their leg. Segmental QFlow hemodynamic and morphological examinations were performed in 30 patients before their superficial venous intervention.

Noninvasive color Doppler ultrasonography (US) and TRANCE-MRI were performed in all 30 patients to assess the venous status of their lower extremities before the scheduled superficial venous intervention (Figure 1). Doppler US was performed in the supine position. The femoral vein (FV), great saphenous vein (GSV), popliteal vein (PV), and perforating vein in the calves were examined. Pelvic veins were not evaluated in the Doppler examination. For further QFlow comparisons, we included 10 healthy volunteers in this study.

**Upper column** (TRANCE MRI venous map):Left great saphenous veinAccessory saphenous veinMajor communicating tributaries.

**Lower column** (Surgical photos)

Primary truncal ablation of the great saphenous vein.Additional ablation of the accessory saphenous vein.Phlebectomy though the small incisions (red arrow).

### 2.2. MRI Acquisition

MRI was performed using a 1.5-T MRI scanner (Philips Ingenia, Philips Healthcare, Best, the Netherlands). The process was carried out with the patients in supine; a peripheral pulse unit trigger was used. The arterial system images were evaluated through a three-dimensional (3D) turbo spin-echo (TSE) skill during systole and diastole periods. TSE TRANCE imaging was conducted using the following parameters: repetition time (TR), 1 beat; echo time (TE), shortest; flip angle, 90°; voxel size, 1.7 mm × 1.7 mm × 3 mm; and field of view (FOV), 350 × 420. The relatively fast arterial blood flow during systole can cause signal dephasing and lead to flow voids. Accordingly, when systolic triggering is applied, the arteries appear black. The relatively slow arterial blood flow during diastole does not cause signal dephasing. Hence, the arteries appear bright on diastolic scans. Subtracting the two phased scans yields a 3D data set of the arteries only. Other images of the venous systems are evaluated through 3D TSE short-tau inversion recovery (STIR) during the systole period. TSE STIR TRANCE imaging was performed using the following parameters: TR, 1 beat; TE, 85; inversion recovery delay time, 160; voxel size, 1.7 mm × 1.7 mm × 4 mm; and FOV, 360 × 320. STIR gives additional background suppression by suppressing connective tissues. When systolic triggering is applied, the arteries appear black. The imaging process yields a 3D data set of the venous system, and no subtraction is required for the data set. A quantitative flow scan is routinely performed to determine appropriate trigger delay times for systolic and diastolic triggering. All images were acquired without the use of gadolinium contrast medium. A QFlow scan entails several acquisitions occurring within one cardiac cycle, resulting in multiple phases. QFlow analysis provides information regarding stroke volume (SV), forward and backward flow volumes, flux, stroke distance (SD), mean velocity (MV), and vessel area. In this study, the postprocessing package calculated quantitative information such as flow velocity, visualized as two-dimensional flow maps overlaid on anatomical references. The bilateral external iliac veins (EIVs), FVs, PVs, and GSVs were analyzed.

### 2.3. Statistical Analysis

Continuous variables (age and QFlow) were analyzed using an unpaired two-tailed Student’s t test or one-way analysis of variance, and discrete variables (sex, substance usage, comorbidities, and intervention history) were compared using a two-tailed Fisher’s exact test. All statistical analyses were conducted using STATA statistics/Data Analysis 8.0 (Stata Corporation, College Station, TX, USA).

## 3. Results

Table 1 summarizes the 30 patients including sex, age, comorbidities, dominant symptoms, target leg, Clinical-Etiology-Anatomy-Pathophysiology (CEAP) classification, and wound location. The mean age of the patients was 58.67 ± 11.98 years, and the majority of the patients were women (24/30, 80%). The dominant symptoms requiring surgical consultation were claudication (13/30, 43%), calf cramping (5/30, 17%), and static leg ulcers (12/30, 40%). The left leg was the target leg requiring treatment in most patients (20/30, 67%). Moreover, most patients were scheduled to undergo left superficial venous interventions, and all patients had lesions over C4. All patients exhibited saphenofemoral junction insufficiency in the Duplex study and morphological features of varicose veins in preoperative TRANCE-MRI.

Table 2 summarizes the interventional data of the 30 patients. In addition to the standard truncal ablation of the GSV, six patients (6/30, 20%) received a second ablation according to TRANCE-MRI mapping for the accessory saphenous vein, posterior accessory saphenous vein (vein of Giacomini), small saphenous veins, and bifurcated GSVs. Varicose veins over 1 cm in diameter on TRANCE-MRI were removed by creating a small incision; veins less than 1 cm in diameter were treated through sclerotherapy.

### 3.1. Comparison between Duplex Scanning and TRANCE-MRI Preoperatively

All 30 patients underwent preoperative duplex scanning and TRANCE-MRI preoperatively. TRANCE-MRI and Duplex identically excluded patients with DVT (Table 3). The TRANCE-MRI criterion of superficial venous reflux (QFlow mean flux (MF) ratio of GSV/PV > 1) was compared with the duplex scan (the gold standard to examine the saphenous femoral junction reflux) with regard to their abilities to detect superficial venous reflux by using a Cohen’s kappa coefficient of 0.967 [10].

### 3.2. Comparison of TRANCE-MRI Hemodynamic Parameters between the Morbid Limbs and Healthy Volunteers

QFlow analysis performed through TRANCE-MRI examined the SV (mL), forward flow volume (FFV, mL), MF (mL), SD (cm), and MV (cm) in the vena cava, EIVs, FVs, PVs, and GSVs in the 30 patients and healthy controls. To decrease bias in the QFlow analysis, we analyzed the same side of the legs of controls as that of patients who received interventions. Table 4 shows the findings of the QFlow comparison of 10 patients who received interventions in their left leg with the left legs of 20 controls. SVs were higher in the EIVs (*p* = 0.021) in the left-leg-intervention group. The MF was higher in the EIV (*p* = 0.012) and tended to increase in the GSV (*p* = 0.087) in the left-leg-intervention group. SD was longer in EIV segments.

Table 5 shows the findings of the QFlow comparison of 10 patients who received intervention in their right leg with the right legs of 10 controls. SV (*p* = 0.002), FFV (*p* = 0.001), and MF (*p* = 0.001) in the GSV were higher in the right-leg-intervention group.

## 4. Discussion

Superficial venous interventions for varicose veins in the lower extremities mainly include truncal ablation, phlebectomy, and sclerotherapy. Patients suspected to have venous reflux disease of the legs undergo air plethysmography and US (duplex) as the initiation of their therapy. US, a rapid tool, can provide additional information regarding active and gravitational refluxes in the standing position when performed by experienced operators. However, US is operator dependent and does not gain information regarding the pelvis. In many institutions, including ours, duplex scanning is exclusively performed in US centers and not performed by the same physician in the clinic; this requires additional communication between staff to gain sufficient surgical information. Meanwhile, pelvic status, including vessel compression and occult benign and malignant pathology, can only be excluded by using other objective diagnostic tools. Venography is historically considered the gold standard for the detection of DVT and other venous occlusive diseases. However, venography is an invasive procedure and cannot reveal varicose veins outside the drainage course of the contrast-medium injection site; thus, it is no longer included in the preoperative evaluation of superficial venous interventions. Intravenous US (IVUS) is an imaging tool used for diagnosing deep vein disease and is mostly used for guiding effective endovascular treatment in iliac and caval venous obstructive diseases [11,12]. However, IVUS is invasive and provides only the details inside the venous lumen without those of the superficial venous system. CT venography may be feasible for the exclusion of pulmonary embolism in patients with symptoms of DVT in the legs; however, CT venography still requires the injection of contrast medium into the morbid limb to achieve optimal venous imaging of the extremities; this procedure can harm the diseased limb [13].

Magnetic resonance angiography (MRA) techniques used for reconstructing vascular structures include time-of-flight (TOF), phase-contrast and electrocardiography (ECG)-gated TSE MRA [11]. The major disadvantages of TOF-MRV are that the FOV is small for each image obtained and that it requires extraordinary time to gain a whole image of the legs. MRI with gadolinium-based contrast medium is a relatively rapid method for imaging the lower extremities [12,14]. Although MRI does not involve radiation exposure, noniodinated contrast agents used in the imaging process still produce undesirable effects. For example, nephrogenic sclerosing fibrosis is a severe complication of gadolinium-based contrast agents in patients with impairment of kidney function and may even occur in patients with normal renal function [15,16]. Phase-contrast MRI depends on phase shifts caused by blood flow. Thus, this technique permits the use of coronal or sagittal slice orientations with an FOV along the direction of the vessel of interest and can quantitatively measure the dynamic flow of the chosen region of interest. Most studies have used phase-contrast MRA for evaluating central nervous system pathologies including hydrocephalus [17,18]. An ECG-gated multistep TSE technique (i.e., TRANCE-MRI) enables the imaging of vessels in the whole lower extremity. ECG gating helps to adapt imaging times to different flow characteristics and therefore optimize image quality faster. Although some studies have used non-contrast-enhanced MRI, most have used this technique to evaluate arterial diseases [19,20,21,22,23]. Our team has innovated the use of TRANCE-MRI to provide more valuable information for the management of complicated lower venous diseases since 2017 [4,5,6,9]. The examination time could be shortened to less than 25 min by our experienced radiological teams with a reasonable cost (250 USD/each exam). The morphology of the venous anatomy of the lower extremities, especially the low-flow superficial venous system, could be clearly demonstrated through 3D imaging without the use of contrast medium or radiation. TRANCE-MRI has been the standard preoperative evaluation modality for superficial venous interventions in our institution and has positive feedback from the patients during the preoperative communication (Figure 2). We use duplex scanning to identify venous thrombosis and superficial venous reflux. TRANCE-MRI can be arranged to plan for further venous interventions. The pelvic status was proven no coexisted external compression inside the pelvis first. The morphology of the GSVs, accessory saphenous veins, and small saphenous veins is routinely examined and referred for anesthesia management in accordance with surgical plans (Appendix A) [11]. TRANCE-MRI reveals the tributaries of the calves in detail, enabling excellent communication with patients with regard to their treatment options such as sclerotherapy and phlebectomy. Our previous study indicated that a TRANCE-MRI GSV/PV ratio of >1 may be a hallmark of superficial venous reflux [10]. These signs were correlated to the duplex findings of these 30 patients.

In this study, we included 10 healthy controls and compared the QFflow analysis findings of the patients who received TRANCE-MRI-guided interventions in their left leg with the left legs of the controls. SV was higher in the EIV (*p* = 0.021) in the left-leg-intervention group. The MF was higher in the EIV (*p* = 0.012) and tended to increase in the GSV (*p* = 0.087) in the left-leg-intervention group. SD was longer in the EIV segments (*p* = 0.019). The QFlow of 10 patients who received interventions in their right leg was analyzed, and SV (*p* = 0.002), FFV (*p* = 0.001), and MF (*p* = 0.001) in the GSV were higher in the right-leg-intervention group. We suppose that the patients with venous reflux who were willing to receive intervention had higher SV and MF in the right legs. Moreover, the left-leg-intervention group exhibited higher left EIV flow, implying that the association of pelvic flow, such as pelvic congestions, may be considered in the left legs.

### Study Limitations

The major limitations of this study are its nonrandomized design and small sample size. This TRANCE MRI-guided superficial venous intervention is a new protocol, thus its impact on the clinical outcome is not available yet. However, this is the first series to discuss the use of TRANCE-MRI in conjunction with superficial venous intervention of the legs. In addition to prove the morphological advantage and safety of TRANCE-MRI, this study analyzed QFlow data in surgical scenarios.

## 5. Conclusions

TRANCE-MRI is useful for excluding pelvic lesions, understanding the truncal anatomy, and localizing the major tributaries in the lower extremities. The QFlow data shows that the MF in the GSVs tended to increase in the patients scheduled for surgical intervention compared with the healthy controls. The reversed GSV/PV ratio in the MF could be observed in the QFlow of the morbid limbs. This promising tool may improve the strategy of superficial venous interventions in the lower extremities.

## 6. Patents

This project is under the reviewing process in the Taiwan Intellectual property Office. (No 109126307).

## Figures and Tables

**Figure 1 jpm-11-00751-f001:**
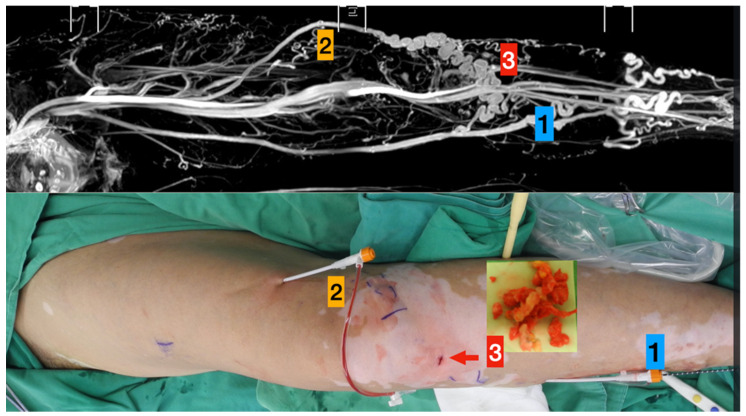
TRANCE MRI-Guiding superficial venous intervention.

**Figure 2 jpm-11-00751-f002:**
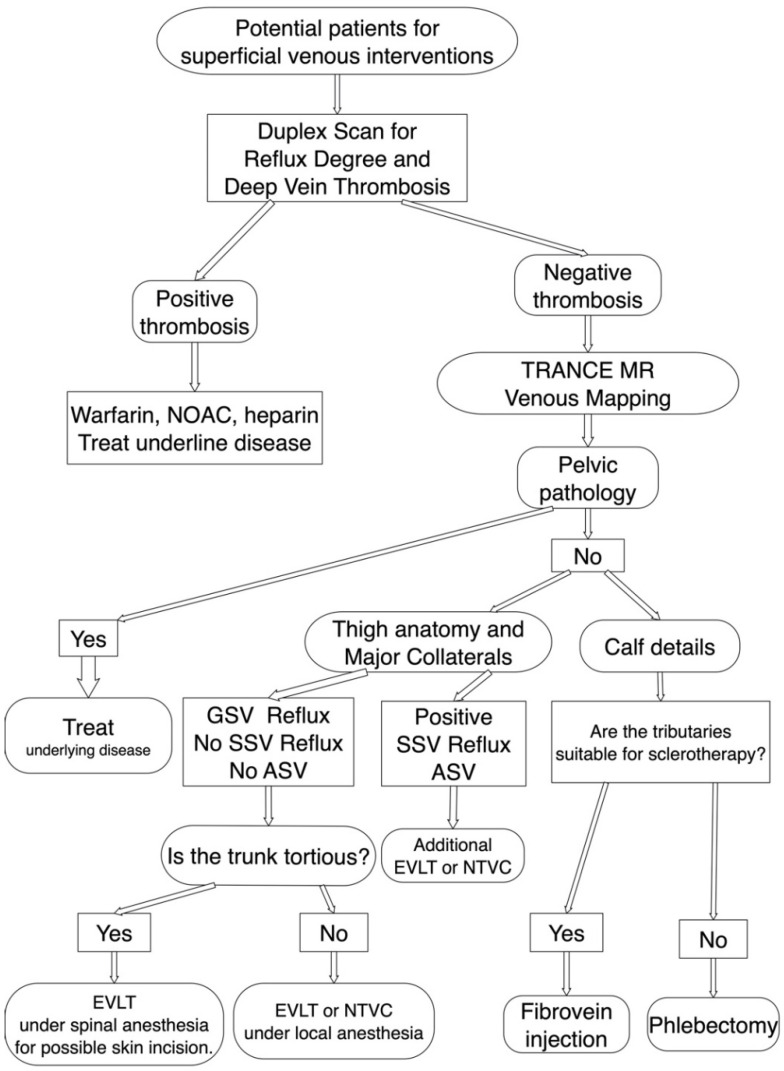
TRANCE MRI venous mapping appliance for preoperative assessment of the superficial venous intervention. Duplex scan was performed first to exclude deep venous thrombosis. If the superficial venous intervention is indicated, we arranged TRANCE MRI mapping to exclude pelvic pathology. Meanwhile, we assess the truncal anatomy, accessory saphenous veins and calf tributaries though TRANCE MRI for complete surgical planning. ASV, accessory saphenous vein; EVLT, Endo-venous Laser Treatment; GSV, great saphenous vein; NOAC, non-coumadin oral anticoagulant; NTVC, non-thermal venous closure; SSV, short saphenous vein.

**Table 1 jpm-11-00751-t001:** Demographic data of the 30 patients with symptomatic varicose vein TRANCE MR as preoperative evaluation.

No	Age	Sex	Comorbidities	Treating Legs	Symptoms	C in CEAP	E in CEAP	A in CEAP	P in CEAP	Wound Location
1	46	F	Nil	Left	Claudication	C4b	Ep	GSVa, GSVb, ASV	Pr	no
2	46	F	Nil	Left	Claudication	C5	Ep	GSVa, GSVb	Pr	no
3	58	F	Nil	Left	Claudication	C4a	Ep	GSVa, GSVb	Pr	no
4	82	F	Nil	Left	Claudication	C4c	Ep	GSVa, GSVb, SSV	Pr	no
5	59	F	HTN	Left	Claudication	C4b	Ep	GSVa, GSVb	Pr	no
6	84	F	Severe MR and TR, CHF	Left	calves cramping	C4c	Ep	GSVa, GSVb	Pr	no
7	58	M	Nl	Left	Claudication	C5	Ep	GSVa, GSVb	Pr	no
8	57	F	Nil	Right	Claudication	C5	Ep	GSVa, GSVb	Pr	no
9	65	F	HTN	Left	calves cramping	C5	Ep	GSVa, GSVb, CPV	Pr	no
10	53	F	Nil	Left	calves cramping	C4a	Ep	GSVa, GSVb, CPV	Pr	no
11	56	F	Nil	Right	Wound	C5	Ep	GSVa, GSVb, SSV, CPV	Pr	medial ankle
12	43	M	Nil	Left	Wound	C5	Ep	GSVa, GSVb	Pr	no
13	59	F	Nil	Left	Claudication	C4b	Ep	GSVa, GSVb, SSV, Vein of Giacomini	Pr	no
14	55	F	HTN	Left	Claudication	C4a	Ep	GSVa, GSVb, CPV	Pr	no
15	69	F	HTN	Right	Claudication	C4a	Ep	GSVa, GSVb, CPV	Pr	no
16	67	F	nil	Left	Claudication	C4c	Ep	GSVa, GSVb, SSV, CPV	Pr	no
17	71	F	Nil	Left	Claudication	C4b	Ep	GSVa, GSVb, CPV	Pr	no
18	38	F	Nil	Right	Claudication	C4c	Ep	GSVb, SSV, TPV	Pr	no
19	59	F	DM	Left	Wound	C6	Ep	GSVa, GSVb, SSV, CPV	Pr	gaiter area
20	68	F	Nil	Right	Wound	C6r	Ep	GSVa, GSVb, CPV	Pr	medial ankle
21	58	F	Nil	Right	Wound	C6	Ep	GSVa, GSVb, CPV	Pr	medial ankle
22	40	M	Nil	Left	Wound	C6	Ep	GSVa, GSVb	Pr	medial ankle
23	43	M	Obese	Left	Wound	C6r	Ep	GSVa, GSVb, SSV, CPV	Pr	lateral malleola
24	53	F	Nil	Right	Wound	C6	Ep	GSVa, GSVb, SSV	Pr	medial ankle
25	50	M	Nil	Right	Wound	C6	Ep	GSVa, GSVb	Pr	medial ankle
26	82	F	Nil	Right	Wound	C6r	Ep	GSVa, GSVb	Pr	medial ankle
27	67	M	CVA, HTN, DM	Left	Wound	C6	Ep	GSVa, GSVb, SSV	Pr	medial ankle
28	58	F	Nil	Left	Wound	C6r	Ep	GSVa, GSVb	Pr	medial ankle
29	54	F	Nil	Left	calves cramping	C5	Ep	GSVa, GSVb	Pr	no
30	62	F	Nil	Right	calves cramping	C5	Ep	GSVa, GSVb, SSV	Pr	no

AASV, anterior accessory saphenous vein; CEAP, Clinical-Etiology-Anatomy-Pathophysiology; CHF, congestive heart failure; CPV, calf perforator vein; CVA, cerebral vascular accident; DM, diabetes mellitus; F, female; HTN, hypertension; GSVa, great saphenous vein above knee; GSVb, great saphenous vein below knee; M, male; MR, mitral regurgitation; SSV, short saphenous vein; TPV, thigh perforator vein.

**Table 2 jpm-11-00751-t002:** Interventional data of the 25 patients with symptomatic varicose vein TRANCE MR as preoperative evaluation.

Patient No	Device	Primary Ablation	Secondary Ablation	Thigh Cutdown	Groin Cutdown	Sclerotherapy	Phlebectomy	Tumescent Solution Use	Complication
1	VNUS (metronic)	LGSV	ASV	Yes	Nil	Calf (alcohol)	Calf and knee	Yes	Nil
2	Atoven catheter	LGSV	Nil	Nil	Yes	Calf (Fibrovein)	Calf	Yes	Echymosis
3	Atoven catheter	LGSV	Nil	Yes	Nil	Nil	Nil	Yes	Nil
4	Venaseal	LGSV	SSV	Yes	Nil	Nil	Nil	Nil	Nil
5	Atoven catheter	LGSV	Nil	Yes	Nil	Calf(Fibrovein)	Nil	Yes	Nil
6	Venaseal	LGSV	Nil	Nil	Nil	Nil	Nil	Nil	Nil
7	Atoven catheter	LGSV	Nil	Nil	Nil	Calf(Fibrovein)	Nil	Yes	Nil
8	Venaseal	RGSV	Nil	Yes	Yes	Nil	Nil	Nil	Nil
9	Atoven catheter	LGSV	LSV	Nil	Nil	Calf(Fibrovein)	Nil	Nil	Nil
10	A.R.C catheter	LGSV	Nil	Nil	Nil	Calf(Fibrovein)	popliteal fossa	Yes	Nil
11	Atoven catheter	LGSV	Nil	Nil	Nil	Calf(Fibrovein)	Nil	Yes	Nil
12	A.R.C catheter	LGSV	Nil	Nil	Nil	Calf(Fibrovein)	Nil	Yes	Nil
13	A.R.C catheter	LGSV	SSV and PASV	Nil	Nil	Calf(Fibrovein)	Nil	Yes	Nil
14	Venaseal	LGSV	Nil	Nil	Nil	Nil	Nil	Nil	Nil
15	Atoven catheter	RGSV	Nil	Nil	Nil	Calf (alcohol)	Nil	Yes	Nil
16	Venaseal	LGSV	Nil	Nil	Nil	Calf(Fibrovein)	Nil	Nil	Nil
17	A.R.C catheter	LGSV	bifurcated GSV	Nil	Nil	Nil	Nil	Yes	Nil
18	A.R.C catheter	RGSV	Nil	Nil	Nil	lateral thigh(Fibrovein)	Nil	Yes	Nil
19	Atoven catheter	LGSV	Nil	Yes	Nil	Calf(Fibrovein)	Calf and knee	Yes	Nil
20	Venaseal	RGSV	Nil	Nil	Nil	Nil	Nil	Nil	topical allergy
21	Atoven catheter	RGSV	bifurcated GSV	Nil	Nil	Calf(Fibrovein)	Nil	Yes	Nil
22	Venaseal	LGSV	Nil	Nil	Nil	Nil	Nil	Nil	Nil
23	A.R.C catheter	LGSV	LSSV	Nil	Nil	Calf(Fibrovein)	Nil	Yes	Nil
24	A.R.C catheter	RGSV	Nil	Nil	Nil	Calf(Fibrovein)	Nil	Yes	Nil
25	A.R.C catheter	RGSV	Nil	Nil	Nil	Calf(Fibrovein)	Nil	Yes	Nil
26	Venaseal	RGSV	Nil	Nil	Nil	Nil	Nil	Nil	Nil
27	A.R.C catheter	LGSV	SSV	Nil	Nil	Calf(Fibrovein)	Nil	Yes	Nil
28	A.R.C catheter	LGSV	Nil	Nil	Nil	Nil	Nil	Yes	Nil
29	A.R.C catheter	LGSV	Nil	Nil	Nil			Yes	Nil
30	Atoven catheter	RGSV	Nil	Nil	Nil	Nil	Nil	Yes	Nil

LGSV: left saphenous vein; RGSV: right saphenous vein.

**Table 3 jpm-11-00751-t003:** Diagnostic tool performance for venous disease in these 30 patients.

No	Dopplex-DVT	Dupplex: SFJ Reflux	Dupplex: Additional Target for Ablation	TRANCE-DVT	TRANCE MR GSV/PV MF QFlow >1	TRANCE-MTS Like Lesion	TRANCE-Additional Target for Ablation
1	No	Yes	No	No	Yes	No	Yes (ASV)
2	No	Yes	No	No	Yes	No	No
3	No	Yes	No	No	Yes	No	No
4	No	Yes	No	No	Yes	No	SSV
5	No	Yes	No	No	Yes	No	No
6	No	Yes	No	No	Yes	No	No
7	No	Yes	No	No	Yes	No	No
8	No	Yes	No	No	Yes	No	No
9	No	Yes	No	No	Yes	No	LSV
10	No	Yes	No	No	Yes	No	No
11	No	Yes	No	No	Yes	No	No
12	No	Yes	No	No	Yes	No	No
13	No	Yes	No	No	Yes	No	Yes (SSV and PASV)
14	No	Yes	No	No	Yes	No	No
15	No	Yes	No	No	Yes	No	No
16	No	Yes	No	No	Yes	No	No
17	No	Yes	No	No	Yes	No	Yes (bifurcated GSV)
18	No	No	No	No	Yes	No	No
19	No	Yes	No	No	Yes	No	No
20	No	Yes	No	No	Yes	No	No
21	No	Yes	No	No	Yes	No	Yes (bifurcated GSV)
22	No	Yes	No	No	Yes	No	No
23	No	Yes	No	No	Yes	Yes	Yes (LSSV)
24	No	Yes	No	No	Yes	No	No
25	No	Yes	No	No	Yes	No	No
26	No	Yes	No	No	Yes	No	No
27	No	Yes	No	No	Yes	Yes	Yes (LSSV)
28	No	Yes	No	No	Yes	No	No
29	No	Yes	No	No	Yes	No	No
30	No	Yes	No	No	Yes	No	No

ASV, accessory saphneous vein; DVT, deep venou thrombosis; LSSV, left short saphenous vein; MF, mean flux; MTS, May-Thurner Syndrome; PASV, posterior accessory saphenous vein; SFJ, sphano-femoral junction.

**Table 4 jpm-11-00751-t004:** Comparison of the QFlow parameters between the left legs of health controls and the left legs planning for truncal ablation.

0		Health Volunteers (N = 10)		Planned Superficial Intervention (N = 20)			Power Analysis		
QFlow	Segments	Mean	Standard Deviation	Mean	Standard Deviation	*p* Value	Power	Effect Size d	Total Sample Size
SV (Stroke Volumes)									
	IVC	18.538	6.125	16.147	6.135	0.349	0.458	0.390	186
	LEIV	3.691	1.050	5.056	1.838	0.021 *	0.768	0.912	36
	LFV	1.202	0.746	1.838	1.417	0.202	0.573	0.561	90
	LGSV	0.459	0.324	1.063	1.145	0.118	0.668	0.718	56
	LPV	0.643	0.332	1.112	1.324	0.285	0.524		
FFV (Foreward Flow Volumes)									
	IVC	18.992	6.192	16.887	6.110	0.410	0.524	0.486	120
	LEIV	3.849	1.114	5.433	2.688	0.090	0.697	0.770	50
	LFV	1.230	0.719	1.855	1.472	0.223	0.559	0.539	98
	LGSV	0.473	0.308	0.834	0.662	0.119	0.657	0.698	60
	LPV	0.654	0.317	1.144	1.409	0.293	0.519	0.479	124
BFV (Backward Flow Volumes)									
	IVC	0.452	1.028	0.738	2.243	0.711	0.303	0.164	1038
	LEIV	0.155	0.240	0.375	1.412	0.632	0.339	0.218	590
	LFV	0.027	0.049	0.017	0.061	0.644	0.321	0.191	764
	LGSV	0.012	0.022	0.333	1.319	0.453	0.393	0.344	212
	LPV	0.009	0.020	0.030	0.095	0.502	0.370	0.306	266
RF (Regurgitant Fraction)									
	IVC	2.186	4.892	3.963	13.140	0.688	0.426	0.344	238
	LEIV	3.749	7.477	3.230	10.679	0.888	0.233	0.056	8780
	LFV	5.206	5.852	0.629	1.877	0.168	0.826	1.053	28
	LGSV	9.650	26.954	3.924	7.372	0.296	0.389	0.290	334
	Lt PV	5.986	8.540	5.291	17.618	0.917	0.230	0.050	11048
ASV (Absolute Stroke Volumes)									
	IVC	19.448	6.426	17.627	6.864	0.512	0.378	0.274	374
	LEIV	4.008	1.222	5.808	3.881	0.169	0.614	0.626	74
	LFV	1.262	0.695	1.872	1.528	0.247	0.543	0.514	108
	LGSV	0.487	0.294	1.169	1.454	0.158	0.629	0.651	68
	LPV	0.665	0.303	1.174	1.495	0.301	0.514	0.472	128
MF (Mean Flux)									
	IVC	21.336	6.848	18.679	7.912	0.395	0.437	0.359	218
	LEIV	3.798	0.871	5.395	2.136	0.012 *	0.797	0.979	32
	LFV	1.246	0.776	1.924	1.268	0.140	0.626	0.645	70
	LGSV	0.477	0.362	1.097	1.057	0.087	0.705	0.784	48
	LPV	0.650	0.322	1.140	1.184	0.215	0.576	0.565	90
SD (Stroke Distance)									
	IVC	9.519	3.317	11.026	5.368	0.438	0.422	0.338	246
	LEIV	3.459	0.590	6.705	5.122	0.019 *	0.758	0.890	38
	LFV	4.092	3.357	4.751	3.008	0.603	0.331	0.207	652
	LGSV	2.005	1.520	2.672	4.919	0.682	0.315	0.183	832
	LPV	1.408	1.124	1.384	0.883	0.952	0.214	0.024	49768
MV (Mean Velocity)									
	IVC	11.169	4.354	12.304	4.975	0.563	0.356	0.243	474
	LEIV	33.816	94.936	7.125	5.424	0.397	0.463	0.397	178
	LFV	4.347	3.893	5.079	3.249	0.604	0.330	0.204	670
	LGSV	2.033	1.554	2.954	4.973	0.577	0.361	0.250	448
	LPV	1.458	1.252	1.452	0.886	0.989	0.203	0.005	>10,000

IVC, inferior vena cava; LEIA, left external iliac vein, LFV, left femoral vein; LGSV, left great saphaneous vein; LPV, left popliteal vein.

**Table 5 jpm-11-00751-t005:** Comparison of the QFlow parameters between the right legs of health controls and the right legs planning for truncal ablation.

		Health Volunteers (N = 10)		Planned Superficial Intervention (N = 10)			Power Analysis		
QFlow	Segments	Mean	Standard Deviation	Mean	Standard Deviation	*p*-Value	Power	Effect Size d	Total Sample Size
SV (Stroke Volumes)									
	IVC	18.538	6.125	13.933	5.537	0.118	0.645	0.789	42
	REIV	4.303	0.872	5.456	2.725	0.282	0.527	0.570	78
	RFV	1.437	0.704	1.764	0.996	0.427	0.414	0.379	174
	RGSV	0.360	0.265	0.893	0.342	0.002 *	0.934	1.742	10
	Rt PV	0.579	0.278	1.125	0.876	0.128	0.670	0.840	38
FFV (Foreward Flow Volumes)									
	IVC	18.992	6.192	14.594	5.395	0.133	0.670	0.840	38
	REIV	4.605	1.074	5.625	2.624	0.278	0.491	0.509	98
	RFV	1.451	0.691	1.774	0.985	0.426	0.414	0.379	174
	RGSV	0.377	0.248	0.903	0.321	0.001 *	0.945	1.832	10
	RPV	0.600	0.277	1.144	0.864	0.125	0.674	0.848	36
BFV (Backward Flow Volumes)									
	IVC	0.452	1.028	0.659	1.102	0.687	0.305	0.194	660
	REIV	0.299	0.396	0.166	0.216	0.380	0.436	0.416	146
	RFV	0.012	0.025	0.009	0.025	0.788	0.267	0.128	1514
	RGSV	0.026	0.046	0.009	0.025	0.351	0.468	0.470	114
	RPV	0.018	0.030	0.016	0.024	0.896			
RF (Regurgitant Fraction)									
	IVC	2.186	4.892	4.755	7.635	0.398	0.232	0.062	6416
	REIV	5.928	7.477	4.024	5.293	0.537	0.363	0.294	288
	RFV	2.317	5.852	1.321	3.178	0.672	0.315	0.211	556
	RGSV	15.705	26.954	2.731	7.725	0.176	0.574	0.654	60
	RPV	4.737	8.540	4.115	6.736	0.869	0.242	0.081	3784
ASV (Absolute Stroke Volumes)									
	IVC	19.448	6.426	15.258	5.474	0.162	0.600	0.702	52
	REIV	4.908	1.367	5.781	2.544	0.364	0.443	0.428	138
	RFV	1.464	0.681	1.784	0.973	0.424	0.415	0.381	172
	RGSV	0.404	0.220	0.913	0.302	0.001	0.955	1.924	10
	RPV	0.620	0.278	1.164	0.852	0.120	0.678	0.858	36
MF (Mean Flux)									
	IVC	21.336	6.848	16.453	7.724	0.174	0.582	0.669	58
	REIV	4.478	0.846	6.363	3.774	0.142	0.593	0.689	54
	RFV	1.471	0.753	2.048	1.293	0.253	0.512	0.545	86
	RGSV	0.372	0.289	1.013	0.400	0.001 *	0.946	1.834	10
	RPV	0.584	0.268	1.348	1.200	0.118	0.688	0.878	34
SD (Stroke Distance)									
	IVC	9.519	3.317	7.770	3.355	0.285	0.500	0.524	92
	REIV	3.862	1.000	5.474	1.619	0.019 *	0.814	1.198	20
	RFV	4.737	3.660	3.026	1.492	0.234	0.550	0.612	68
	RGSV	1.901	1.681	3.156	2.340	0.204	0.553	0.616	68
	Rt PV	1.055	0.450	1.413	1.089	0.357	0.444	0.429	136
MV ( Mean Velocity)									
	IVC	11.169	4.354	9.121	4.894	0.362	0.452	0.442	128
	REIV	4.086	1.246	4.995	2.719	0.359	0.444	0.430	136
	RFV	4.975	4.241	3.433	1.852	0.354	0.469	0.471	114
	RGSV	1.920	1.706	3.475	2.569	0.143	0.605	0.713	52
	RPV	1.064	0.393	1.639	1.362	0.281	0.528	0.573	78

IVC: inferior vena cava; REIA: right external iliac vein; RFV: right femoral vein; RGSV: right great saphaneous vein; RPV: right popliteal vein.

## Data Availability

The data presented in this study are available on request from the corresponding author. The data are not publicly available due to ethical restrictions.

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
