# Peer review of "Superficial Venous Reflux Intervention Guided by Triggered Angiography Non-Contrast-Enhanced Sequence Magnetic Resonance Imaging: Different QFlow Pattern from Health Controls"

_jpm, 2021, doi:10.3390/jpm11080751_

Round 1

Reviewer 1 Report

Dear Authors 

this paper presents some interesting key points and some concerns.

First of all You propose this method for the dianosis and planning of venous insuffiicency, do You think is a simple method to apply?

Did you compare results of patients treated only by US and noted significant difference in term of recurrence?

If You have a non-invasive method (US) with similar results, why we must choice an other more complex?

I agreee with you on the global vision of venous vessel through this MRI , but our aim is clinical outcome for patients. You have to provide a more efficacy of this technique in terms of clinical outcomes. 

Reference: Please correct according to Journal's recommendations

Author Response

Dear reviewers and Editors

Journal: JPM (journal of personalized medicine)

Manuscript ID: jpm-11304540

We are submitting our manuscript entitled “ Superficial venous reflux intervention guided by triggered angiography noncontrast-enhanced sequence magnetic resonance imaging: Different QFlow pattern from Health controls. " for consideration of “Journal of Personalized Medicine” after revise.   Thank you very much again for granting the privilege to us to revise the paper. We have specifically responded to the reviewers’ questions and criticisms point-by-point as follows and add them into this version. Any changes in the manuscript can be tracked by the tool of the MS Word and be marked by underline.

[Comment 1]

You propose this method for the diagnosis and planning of venous insufficiency, do You think it is a simple method to apply?

[Answer 1]

Thank for your thoughtful comment. We totally agree that an Ultrasound though an experienced operator will be more informative, especially in the gravity respond on the standing position. However, the ultrasound machine is not available in the surgical outpatient department in our institution. Moreover, the qualified ultrasound is costy; it needs the most expansive resource: experience operators.

Meanwhile, TRANCE MR did not require contrast media, thus is not very expansive in our health care systems (less than 250 USD/each exam). With the objective venous image (TRANCE MR), the physicians can communicate with the patients in this simple way.

[Change]

We add a short description in discussion, line 214

“The examination time could be shortened to less than 25 min by our experienced radiological teams with a reasonable cost (250 USD/each exam)”

in line 219

“TRANCE-MRI has been the standard preoperative evaluation modality for superficial venous interventions in our institution and has positive feedback from the patients during the preoperative communication”

[Comment 2]

Did you compare results of patients treated only by US and noted significant difference in term of recurrence?

[Answer 2]

Since the TRANCE MR only started since 2017, we have no sufficient data of recurrence after TRANCE-Guide superficial venous interventions.

We have only 5 patients received TRANCE MR after truncal ablation for venous reflux; no residual GSV found yet.

[Change]

No change made

[Comment 3]

If You have a non-invasive method (US) with similar results, why we must choice another more complex?

[Answer 3]

Both US and TRANCE MR are non-invasive manners. Two manners have different advantages: TRANCE MR could see most veins in supine position; however, the calf /short saphenous veins are better checked at standing position by US. We provide both diagnostic options (US with or without TRANCE MR) for those patients query for interventions of their superficial venous reflux in the clinics since 2018.

[Change]

No change made

[Comment 4]

I agree with you on the global vision of venous vessel through this MRI, but our aim is clinical outcome for patients. You have to provide a more efficacy of this technique in terms of clinical outcomes.

[Answer 4]

This technique (TRANCE MR-guided superficial venous intervention) is relative new protocol, thus only short-term outcome is available. We add this weakness into “study limitations”. This TRANCE MR-guided procedure is helpful, especially to the less-experience physicians and the preoperative communication with the patients, although not reflect on the clinical outcome yet.

[Change]

“Study limitations” line 286-287.

“This TRANCE MR-guided superficial venous intervention is a new protocol, thus its impact on the clinical outcome is not available yet.”

Reviewer 2 Report

An excellent study by the authors, MRI can be a very useful tool in superficial venous disease. My only comment is: what are the cost implications and feasbility of this practice in real life? 

Author Response

Reviewer 2

[Comment 1]

An excellent study by the authors, MRI can be a very useful tool in superficial venous disease. My only comment is: what are the cost implications and feasibility of this practice in real life?  

[Answer 1]

We appreciated your encouraging comments for this article. TRANCE MR did not require contrast media, thus is not very expansive in our health care systems (less than 250 USD/each exam). At the beginning of 2017, TRANCE MR exam is very time consuming (one hour operating time for patients and the additional one hour processing time for radiologists/technicians). Now the operating / processing TRANCE MR for venous disease is much reasonable, mostly less than 25 min (including operating and processing time). Most patients could have high image quality, except those patients with extremely arrythmia.

Round 2

Reviewer 1 Report

Dear Authors,

my compliments for  corrections you made. 

Author Response

Thanks for all your comments and encourage.